# Potent Antiviral Activity against HSV-1 and SARS-CoV-2 by Antimicrobial Peptoids

**DOI:** 10.3390/ph14040304

**Published:** 2021-03-31

**Authors:** Gill Diamond, Natalia Molchanova, Claudine Herlan, John A. Fortkort, Jennifer S. Lin, Erika Figgins, Nathen Bopp, Lisa K. Ryan, Donghoon Chung, Robert Scott Adcock, Michael Sherman, Annelise E. Barron

**Affiliations:** 1Department of Oral Immunology and Infectious Diseases, University of Louisville School of Dentistry, Louisville, KY 40202, USA; erika.figgins@louisville.edu; 2Department of Bioengineering, Stanford University School of Medicine, Stanford, CA 94305, USA; nmolchanova@lbl.gov (N.M.); claudine.herlan@kit.edu (C.H.); fortkort@stanford.edu (J.A.F.); jlin3@stanford.edu (J.S.L.); 3Molecular Foundry, Lawrence Berkeley National Laboratory, Berkeley, CA 94720, USA; 4Institute of Organic Chemistry, Karlsruhe Institute of Technology, 76131 Karlsruhe, Germany; 5Department of Pathology, University of Texas Medical Branch, Galveston, TX 77555, USA; nebopp@utmb.edu; 6Division of Infectious Diseases and Global Medicine, Department of Medicine, University of Florida School of Medicine, Gainesville, FL 32601, USA; lisa.ryan@medicine.ufl.edu; 7Center for Predictive Medicine, Department of Microbiology, School of Medicine, University of Louisville, Louisville, KY 40202, USA; hoon.chung@louisville.edu (D.C.); scott.adcock@louisville.edu (R.S.A.); 8Department of Biochemistry and Molecular Biology, Sealy Center for Structural Biology and Molecular Biophysics, University of Texas Medical Branch, Galveston, TX 77555, USA; mbsherma@utmb.edu

**Keywords:** antivirals, peptoids, LL-37, air-liquid interface, cytotoxicity, membrane disruption, COVID-19, HSV-1, SARS-CoV-2

## Abstract

Viral infections, such as those caused by Herpes Simplex Virus-1 (HSV-1) and SARS-CoV-2, affect millions of people each year. However, there are few antiviral drugs that can effectively treat these infections. The standard approach in the development of antiviral drugs involves the identification of a unique viral target, followed by the design of an agent that addresses that target. Antimicrobial peptides (AMPs) represent a novel source of potential antiviral drugs. AMPs have been shown to inactivate numerous different enveloped viruses through the disruption of their viral envelopes. However, the clinical development of AMPs as antimicrobial therapeutics has been hampered by a number of factors, especially their enzymatically labile structure as peptides. We have examined the antiviral potential of peptoid mimics of AMPs (sequence-specific *N*-substituted glycine oligomers). These peptoids have the distinct advantage of being insensitive to proteases, and also exhibit increased bioavailability and stability. Our results demonstrate that several peptoids exhibit potent in vitro antiviral activity against both HSV-1 and SARS-CoV-2 when incubated prior to infection. In other words, they have a direct effect on the viral structure, which appears to render the viral particles non-infective. Visualization by cryo-EM shows viral envelope disruption similar to what has been observed with AMP activity against other viruses. Furthermore, we observed no cytotoxicity against primary cultures of oral epithelial cells. These results suggest a common or biomimetic mechanism, possibly due to the differences between the phospholipid head group makeup of viral envelopes and host cell membranes, thus underscoring the potential of this class of molecules as safe and effective broad-spectrum antiviral agents. We discuss how and why differing molecular features between 10 peptoid candidates may affect both antiviral activity and selectivity.

## 1. Introduction

Herpes simplex virus type-1 (HSV-1) infections cause recurrent oral lesions in the developed world, and are also the primary cause of infectious blindness and genital infections in developed countries. HSV-1 infections also can be life-threatening in immunocompromised individuals [1]. Furthermore, there is recent evidence that HSV-1 infections are associated with the pathogenesis of Alzheimer’s disease [2]. The HSV-1 virus is transmitted readily through oral secretions. It is estimated that 40–80% of the population is infected with this agent, depending on age and socioeconomic status [3]. The primary class of antiviral therapeutics for this pathogen are nucleoside analogues, such as acyclovir or its pro-drug valacyclovir. While acyclovir treatment can reduce the symptoms and shorten the duration of the lesions, it is only effective when given orally, and only reduces the frequency of lesions by approximately 50% [4]. In addition, there is evidence of the development of resistance to this class of antivirals, especially in immunocompromised individuals [5]. Thus, the development of new, effective antiviral agents, which can be used topically to inactivate the virus, is a necessity.

The innate immune system is one of the primary mechanisms for recognizing and eliminating pathogens from mucosal surfaces (reviewed in [6]). Antimicrobial peptides (AMPs) represent an important component of this defense. AMPs are ubiquitous, integral components of innate immunity across phyla, and are promising leads for new antiviral therapies [7] through a number of possible mechanisms. For example, we have recently shown that the human cathelicidin AMP, LL-37, is virucidal for Kaposi’s Sarcoma-associated Herpes Virus (KSHV), through a mechanism by which the cationic peptide disrupts the viral envelope [8].

While many AMPs are potent and selective antibiotics, the typically poor bioavailability of peptides limits their clinical use primarily to topical applications. As a result, only a few AMPs are currently being targeted for systemic delivery [9,10]. Additionally, despite advances in chemical synthesis, manufacturing costs of peptides are still higher in comparison to small-molecule drugs [9,11]. Moreover, while it is possible to improve the protease resistance of AMPs (while maintaining their activity) by designing them with all D-amino acids, this approach significantly increases the cost of peptide synthesis [12,13]. 

The challenges of using AMPs as therapeutics has spurred the development of non-natural peptidomimetics such as ‘peptoids’, which are sequence-specific *N*-substituted glycine oligomers [14,15]. Peptoids are isomerically related to peptides in that their side chains are appended to the backbone amide nitrogens rather than to backbone α-carbons. As a result, peptoids are not proteolyzed by proteases that may be present in the host environment. Furthermore, compared to their peptide counterparts, peptoids have improved biostability and bioavailability, and reduced immunogenicity [16,17]. Peptoid monomer sequences can be designed to form highly stable, amphipathic helices that do not denature. In particular, unlike the helices of peptides which are stabilized by labile hydrogen bonds, the helices of peptoids are stabilized mostly by steric, van der Waal’s and other electronic forces [18,19,20].

Another major advantage of peptoids is that they can be readily synthesized on a peptide synthesizer (typically using a commercially available Rink™ Amide resin), and can be cleaved by trifluoroacetic acid and purified by HPLC. The stepwise “sub-monomer” procedure of adding peptoid repeat units is simpler and less costly than that the corresponding synthesis of peptides [21]. This procedure can be programmed into an automated peptide synthesizer, allowing easy introduction of a variety of side chains that can mimic the structures of most of the 20 natural amino acids, as well as allow the easy introduction of a virtually endles variety of novel side chains [22]. Antimicrobial peptoids have been shown to be analogous to AMPs in terms of their efficacy and structure-activity relationships, thus, suggesting that the two classes of antimicrobial compounds operate via analogous mechanisms [14,23,24,25]. Furthermore, they are generally non-immunogenic, requiring the attachment of a carrier protein to produce an antibody response [26]. Thus, antimicrobial peptoids have significant potential to be developed as a novel class of biostable, peptidomimetic drugs with advantageous properties.

Based on the results demonstrating the inactivation of enveloped viruses with AMPs through a membrane-disruption mechanism, we hypothesized that antimicrobial peptoids would inactivate an enveloped virus such as HSV-1. In addition, since the recently emerged virus, SARS-CoV-2 (the etiologic agent of COVID-19), is similarly enveloped, we further hypothesized that these peptoids would exhibit similar activity against this devastating virus.

Peptoid 1 [H-(*N*Lys-*N*spe-*N*spe)_4_-NH_2_], designated here as MXB-1, is a 12-mer with the trimer sequence motif *N*Lys-*N*spe-*N*spe repeated four times (structures of the monomers are shown in Figure 1A). It is a helical peptoid with broad-spectrum activity against a variety of pathogens [25,27], including viruses as shown here; it is also active against fungi [28] and bacterial biofilms [29]. Despite these promising features, early in vitro studies raised some concerns about the apparent cytotoxicity of Peptoid 1 (notably, however, Peptoid 1 was found to be reasonably well tolerated when delivered intraperitoneally to treat a bacterial infection [25]). We undertook the development of novel analogs of Peptoid 1 that could exhibit reduced cytotoxicity, without compromising the broad-spectrum antimicrobial characteristics of the compound. 

After studying more than 120 peptoid sequence variants, a number of unique peptoids were identified that exhibited potent, broad-spectrum antibacterial in vitro activity. Like the human antibacterial peptide LL-37 and MXB-1 itself, these peptoids induce a rapid rigidification of bacterial cytoplasm as their biomimetic mechanism of action [30], a remarkable biophysical phenomenon that turns early beliefs about AMP mechanisms of action on their head, and which we are currently exploring more deeply.

This effort led to the development of MXB-5 [H-*N*tridec-*N*Lys-*N*spe-*N*spe-*N*Lys-NH_2_; see Figure 1 for structure), a peptoid with a 13-carbon *N*-terminal alkyl modification, that shares some structural similarities with MXB-1 [31]. MXB-5 was found to be highly active against several pathogens as well as against *Pseudomonas aeruginosa* biofilms [29], and was also found to be significantly less cytotoxic to mammalian cells than MXB-1 [32]. In one in vivo experiment which has been presented orally online [33], mice were infected intratracheally with bioluminescent *Pseudomonas aeruginosa*, and were then treated with MXB-5. The peptoid provided a significant reduction in bacterial loads compared to untreated animals, and was also well tolerated in the lungs of the mice. However, MXB-5 was not active against all of the same pathogens that MXB-1 is active against [2].

Given the partial success of MXB-1 and MXB-5, a further library of compounds was developed by hybridizing the key features of these compounds. The activity and potency of these cationic, amphipathic peptoids was found to be affected by their self-assembly into stable, ellipsoidal micelles or other structures (unpublished results; and [34]).

The other contribution to this library came in the form of halogenated peptoids, which were first developed and explored by Molchanova et al. [34]. They synthesized a library of 36 halogenated analogs of MXB-1. These peptoids contained fluorine, chlorine, bromine and iodine atoms, and varied by length and by the level of halogen substitution in position 4 (the para position) of the phenyl rings. Compared to an inactive peptoid hexamer comprising one-half of Peptoid 1/MXB-1 (*N*Lys*N*spe*N*spe*N*Lys*N*spe*N*spe), some of these compounds exhibited improved antimicrobial activity. 

Molchanova et al. found that short (6 mer), Figure 1 brominated analogues of Peptoid 1 not only exhibited significantly reduced cytotoxicity, but relative to non-brominated compounds, displayed up to a 32-fold increase in activity against *S. aureus* and a 16- to 64-fold increase in activity against *E. coli* and *P. aeruginosa*. They ascribed these results to the relatively increased hydrophobicity and self-assembly properties of the compounds. In particular, they obtained small angle X-ray scattering (SAXS) data which demonstrated how the self-assembled structures are dependent on the size of the halogen, the degree of halogen substitution and the length of the peptoid, and they correlated these features to the activity of the resulting peptoid. Thus, the creation of the peptoid library tested in this study hybridizes the molecular features of three different peptoid motifs: (1) inclusion of the *N*Lys*N*spe*N*spe trimer sequence motif; (2) *N*-terminal alkylation with either a C10 or C13 chain; (3) inclusion of para-brominated (at position 4 in the phenyl ring) *N*spe monomers. Within a library comprising these attributes, we identified numerous active antiviral peptoids.

## 2. Results

We tested the 10 peptoids shown in Figure 1B (as well as the natural human host defense peptide LL-37) for activity against HSV-1. This was accomplished by incubating the virus with the peptoid at 20 µg/mL for 2 h at 37 °C, prior to using the incubated virus to infect cultures of OKF6/TERT-1 cells. An initial screening of the peptoids (Figure 2A) showed wide variability in activity, with MXB-10 exhibiting no inhibitory activity. In a separate study, MXB-6 also showed no activity (data not shown). We chose five peptoids to study further. The results in Figure 2B show a dose-dependence with strong activity at 20 µg/mL for MXB-4, MXB-5 and MXB-9 (corresponding to 15 µM, 21.2 µM and 14.5 µM, respectively). MXB-4 and MXB-5 showed a time-dependent inactivation of HSV-1, with activity seen as early as 30 min. MXB-9 exhibited the most potent activity, with complete inhibition of the virus after as little as 30 min (Figure 2C). Since the virus strain expressed Green Fluorescent Protein (GFP), we could observe fluorescence in the cells at 24 h post-infection, indicating viral replication. The control (medium) treated virus exhibited strong fluorescence, while the cultures infected with the virus treated with MXB-5 showed very weak fluorescence in a small number of fluorescent cells. We were required to increase the exposure significantly in order to observe the minimal fluorescence in the peptoid-treated cultures, which led to a bright background (Figure 2D).

Our previous results demonstrated that LL-37 disrupted the membrane of Kaposi’s Sarcoma Herpes Virus (KSHV) [8]. Thus, we hypothesized that these peptoids, which are simplified structural mimics of natural antimicrobial peptides, would act through a similar biophysical mechanism. To examine this, we treated HSV-1 with peptoids, followed by electron microscope imaging of the peptoid-treated viral particles. Using negative staining EM (Figure 3A) we observed disruption of the viral membrane in all treated samples, compared with virus that was treated with PBS alone. Since negative staining procedures could potentially cause modification of the virus owing to chemical treatment of the sample by a heavy metal (tungsten) salt, we decided to verify the results by using cryo-EM without chemical fixation or negative staining. Both techniques lead to the same conclusion.

Selected examples of cryo-EM images are shown in Figure 3B, with two different representative images in each column. The images show clearly resolved viral envelope in the control panels. In the treated samples, there are images showing both partially disrupted envelopes, as well as numerous capsids without envelopes, suggesting a total disruption of the membrane. Quantification of the images (Figure 3C) show that while a small number (21%) of unenveloped capsids are observed in the untreated controls, we see a large increase in both unenveloped capsids (bottom panels, MXB-5 and MXB-9) and disrupted envelopes (top panels). These results are similar to what we observed with treatment of KSHV with LL-37 [8].

To determine whether the peptoids would exhibit similar activity against other enveloped viruses, we tested the inhibitory activity of MXB-4 and MXB-9 against SARS-CoV-2. When incubated with virus for 1 h at 37 °C at increasing concentrations, we observed antiviral activity with approximate IC_50_ values of 20 µg/mL and 7 µg/mL, respectively (Figure 4). Thus, these two peptoids are active at similar relative concentrations against both viruses.

To determine whether the peptoids acted through a similar membrane-dependent mechanism as with HSV-1, we treated SARS-CoV-2 with the two active peptoids (MXB-4 and MXB-9) followed by visualization by Cryo-EM. While SARS coronaviruses to do not exhibit the well-formed hexagonal capsids observed in HSV-1 [35], we observed numerous partially disrupted envelopes as well as what appear to be unenveloped nucleocapsids (Figure 5). We did not observe these structures in the control samples, suggesting that the peptoids are acting through a similar membrane-disruptive mechanism on both types of virus.

To provide further evidence to support the development of these agents as potential therapeutics, we assayed cytotoxicity against cultured cells. To provide the optimal in vitro toxicity model, we used 3-dimensional primary cultures of oral epithelial cells (EpiOral, MatTek Life Sciences, Ashland, MA, USA), grown at an air-liquid interface. The cultures were tested in the presence of increasing concentrations of MXB-4, MXB-5 and MXB-9 for 3 h, applied to the apical surface. Viability was monitored using the MTT assay. The results in Figure 6 show no observable cytotoxicity as assessed by this metabolic assay, at any concentration up to 256 µg/mL. Incubation with 50% ethanol led to 0% viability (not shown).

## 3. Discussion

Herpesvirus infections are common and easily transmissible. However, due to the mechanisms of infection and pathogenesis at play, it has been difficult to design any effective vaccines to prevent herpesvirus infections. Furthermore, other preventive and palliative methods have been challenging to adapt to broad clinical use. Since AMPs such as LL-37 exhibit potent activity against HSV-1 [36], their potential as antiviral preventive or therapeutic agents is significant. However, direct introduction of these peptides has not translated into effective treatments, due to high cost, molecular instability, and unknown pharmacokinetics, among other problems [37,38]. Here, we have demonstrated the potential use of a novel AMP mimetic structure which, based on its mechanism of action, not only inactivates HSV-1, but can be used against other, potentially more dangerous enveloped viral pathogens such as SARS-CoV-2.

Conventional development of antiviral therapeutics usually is based on designing drugs that target enzymes or other targets specific to each virus. On rare occasions, drugs such as Remdesivir, which was originally designed to treat Ebola virus infections by targeting the RNA-dependent RNA polymerase of the virus, can have some efficacy in COVID-19 [39]. However, in general, this means that new therapeutic agents must be developed for each virus. The advantage of AMPs and antimicrobial peptoids is that they often target membranes, regardless of microbial species. In many cases, AMPs have been shown to disrupt the viral envelope, although other non-membrane dependent mechanisms have also been observed [7]. Confirming the results of Gordon et al. [36], and in contrast to the results shown by Roy et al. [40], we observe direct inactivation of HSV-1 by LL-37 outside of the cell. This inactivation is presumably due to the damage to the viral envelope we observed by EM, which would prevent binding and infection. This is in contrast to the small antiviral molecule, LJ001. This therapeutic intercalates into the viral envelope, thus interfering in membrane fusion, without causing envelope disruption [41]. Thus, we hypothesized that antimicrobial peptoids would similarly inactivate enveloped viruses. Our results clearly show inactivation of HSV-1, with widely variable activity among the 10 different peptoids we studied, which is quite remarkable given many similarities in their structures. Some were completely inactive, e.g., Peptoid MXB-6.

Comparing the structures of the ten peptoids on our new library with regard to antiviral activity, halogenation seems to be beneficial as every active compound (except MXB-5) includes at least two *N*spe(p-Br) submonomers. Interestingly, shortening of the peptoid oligomers seems to be an issue as well. MXB-5 is the only derivative that is not characterized by at least two helix turns, but shows strong activity despite its lack of halogen substituents. HSV-1 inhibition appears strongly structure-dependent, as slight changes result in structure give rise to significantly different activities. For example, while MXB-9 exhibits potent antiviral activity, its structural analog MXB-10 has no effects on viral propagation. This difference in activity potentially can be explained by differences in net molecular hydrophobicity upon halogen substitution as previously reported by Molchanova et al. [34]. In this study they found an increasing antimicrobial activity correlating with increasing hydrophobicity, until a threshold was reached where the activity dropped. Beyond the hydrophobicity, recent Small Angle X-ray Scattering data we have acquired for these peptoids in aqueous media have revealed large differences in the self-assembled structures of the peptoids in aqueous PBS buffer. While MXB-9 and MXB-5 both seem to self-assemble into stable core-shell ellipsoidal micellar structures, MXB-10 forms long worm-like cylinders in solution (unpublished data, manuscript in preparation) which may provide an explanation for differences in activity for these peptoids. Further studies are needed to make definitive conclusions about these effects. We believe that a certain type and degree and stability of peptoid self-assemblies in aqueous media can be an important determinant of both activity, and selectivity. If assemblies remain intact near zwitterionic mammalian cells, but disassemble near anionic pathogenic membranes, they can be selective towards affecting pathogens while causing little to no harm to the host.

Lack of cytotoxicity for the host cell is important for any therapeutic. Since HSV-1 generally affects oral epithelial tissues, we tested the active peptoids on well-differentiated 3D cultures of oral epithelium. The drugs were applied to the apical surface, and showed no cytotoxic effects after a 3-h exposure. This is in line with results routinely obtained with AMPs such as LL-37, which only exhibits cytotoxic effects on host cells at high concentrations [40]. 

This raises a very interesting question regarding the mechanism. Our results strongly suggest that, like known AMP activity against bacteria and fungi, the peptoids target and disrupt pathogenic membranes. For bacteria and fungi, the difference between the microbial membrane and the host cell membrane is striking. Microbial phospholipid bilayers have predominantly anionic headgroups on the outer leaflet, in contrast to mammalian cells where the anionic headgroups are mostly on the inner leaflet. This allows for increased binding of cationic peptides, which leads to the formation of pores or other membrane disrupting events (reviewed in [42]). However, viral envelopes are obtained from the host cell, thus suggesting that peptoids or peptides would have a similar propensity for binding to viral membranes as they do to the host cell membrane. Since neither peptoids nor peptides exhibit cytotoxicity at the same concentrations at which they exhibit viral inhibition, this implies either that the makeup of the outer leaflet of the viral envelope differs from that of the host cell membrane, or that some other structural features of the viral envelope are encouraging binding. It is known that the envelopes of many enveloped viruses contain phospholipid head groups that differ from those in the host cell membrane, including that of HSV-1 [43]. Indeed, they often contain increased levels of phosphatidylserine, a negatively charged headgroup that is usually found in the inner leaflet of the plasma membrane [44]. This switching allows the virus to appear more like an apoptotic cell in order to increase uptake by host cells. Recently, a cationic AMP was shown to exhibit increased binding to a number of enveloped viruses through binding to the PS on the envelopes of these viruses [45].

Other possible features of the viral envelope that could enhance peptoid binding could include the curvature of membrane (which is much stronger in the small viral particle), or an abundance of negatively charged amino acids in the glycoproteins on the surface of the envelope. Due to its much smaller radius, the outer leaflet of the viral particle has an 11% greater surface area than the inner leaflet (as opposed to cells, in which this difference in surface area is only 0.1%) [46]. This feature may enhance interaction with the peptoid. Furthermore, a recent study showed that LL-37 binds to the spike protein of SARS-CoV-2 [47], This suggests that the initial interaction of the peptoid with this virus is through binding to the spike protein, rather than to the membrane itself. Further research is necessary to understand completely how AMPs and antimicrobial peptoids can distinguish between viral envelopes and host cell membranes. However, regardless of the specific molecular mechanism, our results suggest that peptoids of the type described herein could be developed as broad-spectrum antivirals that could target numerous enveloped viruses, including those which are highly pathogenic.

## 4. Materials and Methods

### 4.1. Viral Strains and Antiviral Assays

HSV-1-GFP was propagated in Vero E6 cells, plaque purified and titered as described [48]. Stocks were aliquoted and maintained at 10^8^ pfu/mL in −80 °C. Virus was diluted to 10^5^ pfu/mL, and incubated with peptoids at 37 °C. Samples were removed and added to cultured OKF6/TERT-1 cells at an MOI of 0.1:1 in triplicate, and incubated at 37 °C for 24 h. Cells were lysed and total DNA was isolated using a DNA isolation kit (Qiagen). HSV-1 genomic DNA was quantified by QPCR relative to genomic -actin DNA. Results were presented as fold change compared to control.

SARS-CoV-2 (strain USA-WA1/2020 isolate; BEI resource cat# NR-52281) was obtained from BEI resources and amplified in Vero E6 cells. Amplified stock virus was stored at −80 °C until used. For antiviral activity test, virus was diluted in cell culture media to 2000 pfu/mL and mixed with peptoids 1:1 in volume. The mixture was further incubated at 37 °C for one hour and 250 µL of mixture was added to Vero E6 cells grown in 12-well plates. After one hour of incubation, cells were washed once with PBS and overlayed with Avicel overlay media (1% Avicel in DMEM with 10% FBS). Three days later, the overlay media were removed and cells were stained with crystal violet solution (1% crystal violet, 2% paraformaldehyde, 25% Ethanol) for 4 h. Viral plaques were counted and compared to the mock treated samples. Experiments were performed with three replicates per treatment. 

SARS-CoV-2 (strain USA-WA1/2020 isolate) was provided by the World Reference Center for Emerging Viruses and Arboviruses (WRCEVA). Vero76 cells were infected with an MOI of 3 and the infection allowed to progress for 48 h. Supernatants from infected cells were collected and a primary centrifugation of 10,000× *g* was performed for 30 min. Supernatants from this centrifugation were carefully removed to prevent the disruption of the pelleted debris. Supernatants were then laid upon a 10 mL cushion of 20% sucrose and centrifuged for 3 h at 100,000× *g*. Supernatants and sucrose were removed following virus pelleting and the virus pellet resuspended in PBS for immediate cryo-EM processing.

Vero E6 cells (ATCC CRL-1586) were obtained from ATCC and cultured in DMEM (HyClone. Cat. SH30243.FS) with 10% FBS. The cells were cultured in a 37 °C incubator with 5% CO_2_ and passaged every 7 days.

### 4.2. Cytotoxicity 

Normal human bronchial epithelial cells, obtained from MatTek (EpiAirway) were grown at an air-liquid interface at 37 °C. Serial dilutions of peptoid were performed in triplicate using 400 µM peptoid stocks resulting in final concentrations between 200 µM and 6.25 µM. Peptoids were applied to the apical surface of cultures in 100 µL for three hours. Cell viability was quantified using the CyQUANT MTT Cell Viability Assay (ThermoFisher, Waltham, MA, USA), following the manufacturer’s instructions Absorbance was read at 540 nm and percent survival was calculated relative to untreated cultures. Ethanol was used as a positive control for cytotoxicity.

### 4.3. RNA/DNA Isolation

Cells were harvested and homogenized using Qiagen QIAshredder protocol. The obtained lysate was then used to extract RNA and DNA using the Qiagen RNeasy Plus Micro kit. Once RNA was extracted using the manufacturer protocol, DNA was extracted as described [49]. Using the same RNeasy spin columns 50 µL of 8.0 mM NaOH was added directly to membrane. The tubes were then incubated for 10 min at 55 °C. DNA was eluted by centrifugation for 3 min at 5000× g.

### 4.4. Quantitative Polymerase Chain Reaction 

qPCR was performed on a BIO RAD myCycler using SsoAdvanced Universal SYBR green supermix (Bio-RAD #1725274). Purified HSV1 DNA (1.5 µL) was used as the template for each reaction mixture. Relative HSV-1 DNA levels were quantified using HSV UL30 (F: AGAGGGACATCCAGGACTTTGT; R: CAGGCGCTTGTTGGTGTAC), normalized to β-Actin (F: GGA TCA GCA AGC AGG AGT ATG; R: AGA AAG GGT GTA ACG CCA CTA A) in each sample, and the change in expression was determined using the 2^−ΔΔ*C*t^ method compared to control treated samples. 

### 4.5. General Synthesis of Peptoids

Peptoids were synthesized manually in accordance with the submonomer method [21]. All reaction steps were performed in fritted 10 mL syringes under smooth mixing on a VWR^®^ Tube Rocker at 21 °C. Rink amide MBHA resin (Protein Technologies Inc., Tucson, AZ, USA, 0.64 mmol/g) was used as a solid support. Acetylation steps were carried out for 30 min, substitution for 1 h. Substitution with alkylamines was performed overnight. Acetylation using bromoacetic acid and substitution by various amines were alternated until the desired chain length was achieved. The single oligomers were cleaved and deprotected simultaneously using a cocktail of trifluoroacetic acid/triisopropylsilane/water (95:2.5:2.5 (*v*/*v*)) for 30 min. After purification, exchange of the counterion was carried out using a 10 mm solution of aqueous HCl. Lyophilization yielded the desired compound. Purity was determined to exceed 95%. 

### 4.6. Peptoid Purification

Starting materials and solvents were purchased from commercial suppliers (Acros Organics, Alfa Aesar, Chem-Impex Intl. Inc., CNH Technologies, Merck, OmniSolv, Protein Technologies, Sigma-Aldrich, TCI, and VWR) and used without further purification. Water was filtered through a 0.22-μm Millipore membrane filter. 

Product formation and purity were determined by means of analytical UPLC/MS using a Water Acquity UPLC system, equipped with an Acquity Diode Array UV detector and a Waters SQD2 mass spectrometer As stationary phase, a Waters Acquity UPLC Peptide BEH C18 Column (300 Å pore size, 1.7 µm particle size, 2.1 mm × 100 mm) with an Acquity UPLC BEH C18 VanGuard pre-column (1.7 μm, 2.1 mm × 5 mm) was employed. For alkylated derivatives, a Waters Acquity UPLC BEH300 C4 column (300 Å pore size, 1.7 μm particle size, 2.1 mm × 100 mm) with an Acquity UPLC BEH300 C4 VanGuard pre-column (1.7 μm, 2.1 mm × 5 mm) was used. Elution was performed using an aqueous acetonitrile gradient with 0.1% (*v*/*v*) trifluoroacetic acid added (5–95% acetonitrile (*v*/*v*) over 6.80 min, flow rate: 0.8 mL/min, column temperature: 60 °C). UPLC chromatograms are available in the supporting information (see Appendix A). The mass spectra were collected in ESI^+^ mode.

Purification by means of preparative HPLC was carried out using a Waters Prep150LC system, equipped with a Waters 2489 UV/Visable detector and a Waters Fraction Collector III collector. As stationary phase, a Waters XBridge BEH300 Prep C18 column (5 μm particle size, 19 mm × 100 mm) with a Waters XBridge Peptide BEH300 C18 guard column (5 μm particle size, 19 mm × 10 mm) was employed. For alkylated derivatives, a Waters Symmetry300 C4 column (5 μm particle size, 19 mm × 100 mm) with a Waters Symmetry300 C4 guard column (5 μm particle size, 19 mm × 10 mm) was used. Elution was performed using an aqueous acetonitrile gradient with 0.1% (*v*/*v*) trifluoroacetic acid added (20–60% acetonitrile (*v*/*v*) over 30 min at a flow rate of 17 mL/min).

### 4.7. Electron Microscopy

#### 4.7.1. Negative Staining EM

HSV-1 (1 × 10^8^ pfu/mL) was incubated with peptoids or control medium for 2 h. at 37 °C. After incubation, the virus was fixed by addition of 2 volumes 4% glutaraldehyde/PBS, and fixed at 4 °C for 24 h. Virus was applied to grids (ultrathin carbon with lacey support film, Ted Pella, Inc.) for 1 min, followed by staining in 1% phosphotungstic acid for 1 min. Excess liquid was blotted and grids were allowed to air dry. Images were collected using a Thermo Scientific™ Talos™ F200X at 200 kV accelerating voltage with 50 μm objective aperture inserted to enhance contrast. Images were captured using a 4k × 4k CMOS camera (Thermo Scientific, Waltham, MA, USA, Ceta 16M™).

#### 4.7.2. Cryo-EM

After incubation of HSV-1 virus with peptoids for 2 h. at 37 °C, samples were vitrified using Leica EM-GP2^®^ plunger (Leica Microsystems) as previously described [50] on carbon holey film (R2 × 1 Quantifoil^®^; Micro Tools GmbH, Jena, Germany) grids. Briefly, suspensions of virions were applied to the holey films, blotted with filter paper, and plunged into liquid ethane. 

SARS Co-V-2 was incubated with peptoids similar to the HSV-1 (two hours at 37 °C) and vitrified also on R2 × 1 Quantifoil grids using a manual cryo-plunger [51] in a biosafety cabinet. All experiments, including EM imaging were performed in a biosafety level 3 (BSL-3) containment because of the NIH/CDC classification of the virus.

The grids were imaged in a JEOL 2200FS electron microscope (JEOL, Tokyo, Japan). The microscope was operated at 200 keV; we used 20 eV electron energy filter slit for image acquisition. A Fischione Instruments 2550 cryo-transfer side-entry holder (E.A. Fischione Instruments, Inc., Export, PA, USA) was used for data collection. The image were acquired on DE-20 (Direct Electron, San Diego, CA, USA) camera, used in linear mode with 25 frames/s rate. Total electron dose/image was ~25 electrons/Å^2^. Image pixel size was 1.5 Å on the specimen scale.

## Figures and Tables

**Figure 1 pharmaceuticals-14-00304-f001:**
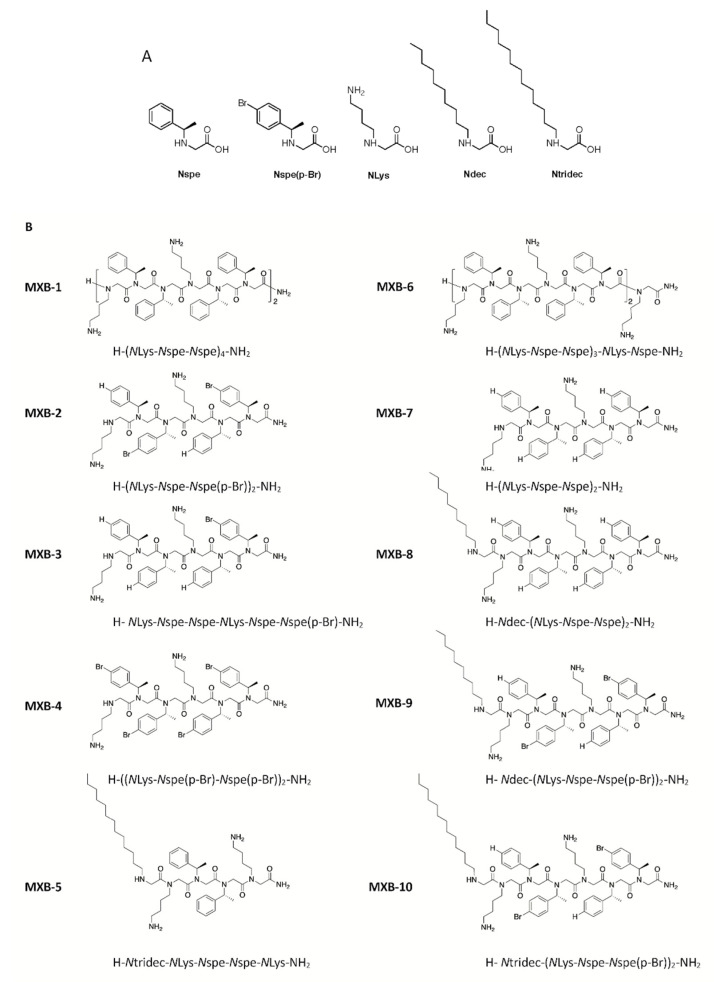
Structure of peptoids. (**A**) Structures of peptoid submonomers used in this library of new compounds. (**B**) Chemical structures of the ten different peptoids used in this study.

**Figure 2 pharmaceuticals-14-00304-f002:**
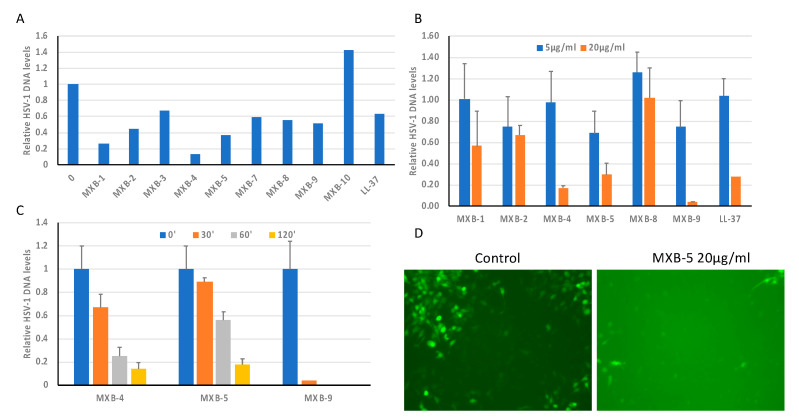
Activity of peptoids and of LL-37 against HSV-1. (**A**) Screening of peptoids against HSV, with comparison to the antiviral activity of LL-37, which is known to be active against HSV-1. The virus was treated with peptoids at 20 µg/mL for 2 h at 37 °C, followed by infection of OKF6/TERT1 cells for 24 h. Total DNA was isolated and HSV-1 DNA was quantified by QPCR relative to β-actin. (**B**) Activity of peptoids MXB-4, MXB-5 and MXB-9 at 5 µg/mL and 20 µg/mL for 2 h. (**C**) Time course of MXB-4, MXB-5 and MXB-9 activity. (**D**) Fluorescence imaging of HSV-1 infection 24 h. after treatment with either control (media) or MXB-5 (20 µg/mL for 2 h). Results shown are representative of at least two independent experiments.

**Figure 3 pharmaceuticals-14-00304-f003:**
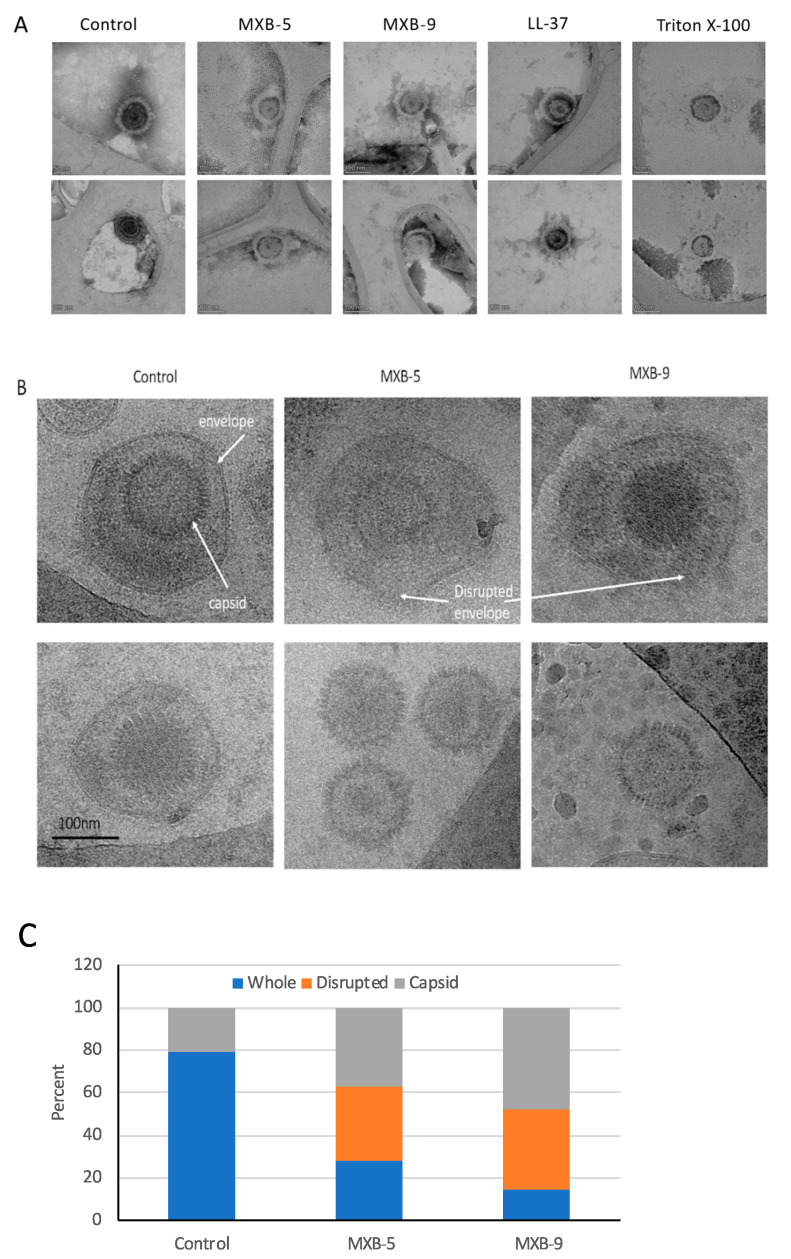
Transmission electron microscope imaging of peptoid-treated HSV-1 vs. controls and comparative analysis of results. Virus was treated for 2 h at 37 °C followed by preparation for EM imaging as described in Materials and Methods. (**A**) Negative stained EM. (**B**) Cryo-EM. Visible in the control samples are the viral envelope and intact capsid (both panels). In the samples treated with the peptoids we observe both disrupted envelopes (top panels) and naked capsids (bottom panels). (**C**) Quantification of relative amounts of whole virus, disrupted envelopes and naked capsids that were observed in the cryo-EM images.

**Figure 4 pharmaceuticals-14-00304-f004:**
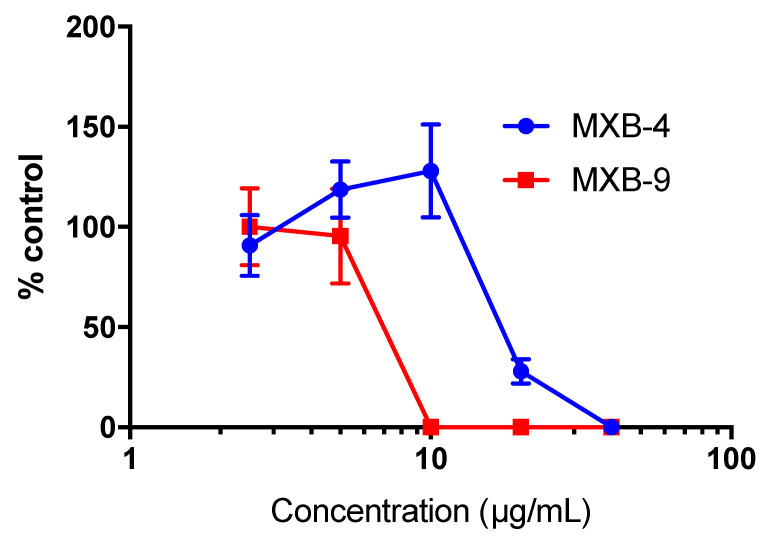
Antiviral activity against SARS-CoV-2. Virus was incubated with peptoids or PBS (control) for 1 h at 37 °C (*n* = 3 per condition), prior to infecting Vero E6 cells. After 3 days, total plaques were counted. Data are expressed as percent of control-treated virus. Results shown are representative of two independent replicate experiments.

**Figure 5 pharmaceuticals-14-00304-f005:**
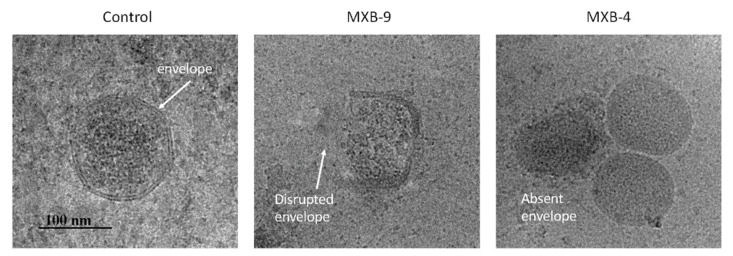
Cryo-EM imaging of peptoid-treated SARS-CoV-2. Virus was treated for 2 h at 37 °C followed by preparation for EM imaging as described in Materials and Methods. Visible are partially disrupted membrane (center) and completely absent membrane (right panel, next to two unaffected virions).

**Figure 6 pharmaceuticals-14-00304-f006:**
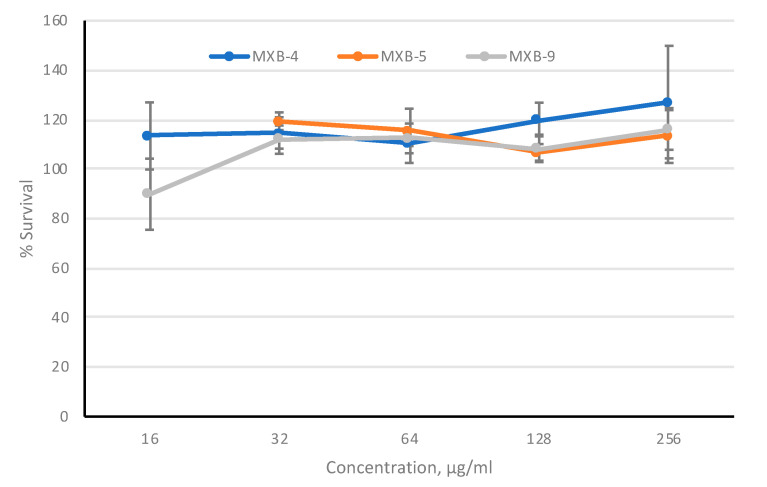
In vitro toxicity of peptoids. Peptoids were incubated on the apical surface of cultured EpiOral cells (*n* = 3 wells per concentration) for 3 h. Cell viability was quantified by MTT assay and are shown as mean percent survival +/− SD. Results shown are representative of two independent replicate experiments.

## Data Availability

The data presented in this study are available on request from the corresponding authors.

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
