# Peer review of "Potent Antiviral Activity against HSV-1 and SARS-CoV-2 by Antimicrobial Peptoids"

_pharmaceuticals, 2021, doi:10.3390/ph14040304_

Round 1

Reviewer 1 Report

The article by Diamond et al. describes the antiviral potential of LL-37 derived peptoids against against HSV-1 and SARS-CoV-2. An in vitro proof-of-concept is described including the antiviral activity by fluorescence and Electron Microscopy together with low cytotoxicity finding by the MTT assay. The paper is well-written and in vitro experimental evidence supports the potential of the compounds.

Hence, the paper is suitable for publication in Pharmaceuticals, but some points need to be addressed before publication.

Comments:

Abstract, Line 25: There is some text missing at the beggining: “Viral infections, such as those caused by Herpes Simplex Virus-1 (HSV-1) and SARS-CoV-2, affect millions of people...”

Line 119-21: Authors say: "We were required to increase the exposure significantly in order to observe any fluorescence in the peptoid-treated cultures, which led to a bright background (Figure 2D)." and figure 2D shows apparently, a bright flurorescence with MXB-5 at 20 mg/mL.

I understand that the objective consists of not detecting the virus fluoresecence because of the antiviral action of the peptoid:

"The control (medium) treated virus exhibited strong fluorescence, while the cultures infected with the virus treated with MXB-5 showed very weak fluorescence in a small number of fluorescent cells..."

Please, authors, could you clarify the above sentence ("increasing exposure to observe fluorescence in peptoid-treated cultures")? what exposure is the text referring to?

Figure 3: EM images have poor resolution, at least in my copy. Could authors improve the quality of the images (and mark, for instance, with small arrows the damaged envelopes and capsids) It will certainly help the reader.

Line 284: heading of section 4.1: Viruses? or better " In vitro antiviral cell culture test", or similar?

Line 343: section 4.5 "Rationale of the design...." is placed in the Materials and Method section. It seems more appropriate to place it before the Results section (or even at the Discussion section, according to the text) as it includes background information and references

Author Response

Abstract, Line 25: There is some text missing at the beginning: “Viral infections, such as those caused by Herpes Simplex Virus-1 (HSV-1) and SARS-CoV-2, affect millions of people...”

Response: I’m not sure what’s missing. This seems ok to me.

Line 119-21: Authors say: "We were required to increase the exposure significantly in order to observe any fluorescence in the peptoid-treated cultures, which led to a bright background (Figure 2D)." and figure 2D shows apparently, a bright fluorescence with MXB-5 at 20 mg/mL.

I understand that the objective consists of not detecting the virus fluorescence because of the antiviral action of the peptoid:

"The control (medium) treated virus exhibited strong fluorescence, while the cultures infected with the virus treated with MXB-5 showed very weak fluorescence in a small number of fluorescent cells..."

Please, authors, could you clarify the above sentence ("increasing exposure to observe fluorescence in peptoid-treated cultures")? what exposure is the text referring to?

            Response: We have amended the text as follows:

“Since the virus strain expressed Green Fluorescent Protein (GFP), we could observe fluorescence in the cells at 24 hours post-infection, indicating viral replication. The control (medium) treated virus exhibited strong fluorescence, while the cultures infected with the virus treated with MXB-5 showed very weak fluorescence in a small number of fluorescent cells. We were required to increase the exposure significantly in order to observe the minimal fluorescence in the peptoid-treated cultures, which led to a bright background (Figure 2D).”

Figure 3: EM images have poor resolution, at least in my copy. Could authors improve the quality of the images (and mark, for instance, with small arrows the damaged envelopes and capsids) It will certainly help the reader.

Response: We have added arrows and descriptions of the envelopes and capsids, and have increased the resolution of the figure.

Line 284: heading of section 4.1: Viruses? or better " In vitro antiviral cell culture test", or similar?

Response: We have changed the heading to “Vial strains and antiviral assays”

Line 343: section 4.5 "Rationale of the design...." is placed in the Materials and Method section. It seems more appropriate to place it before the Results section (or even at the Discussion section, according to the text) as it includes background information and references

Response: This has now been placed at the end of the introduction.

Reviewer 2 Report

The manuscript entitled "Potent antiviral activity against HSV-1 and SARS-CoV-2 by antimicrobial peptoids" (pharmaceuticals-1156528) by Gill Diamond at al. presents the antiviral activity of a set of peptoids able to disrupt the viral membranes of both HSV-1 and SARS-CoV-2. The evidence is shown by activity measurements, fluorescence imaging and transmission electron microscopy. The introduction is well written and the second part of the discussion is particularly insightful. I read the article with great interest. On the other hand I have to admit that the activity measurements on peptoids are somewhat surprising to me. Very small changes in the molecular formula seem to result in dramatic changes in activity. That said, I highlighted some minor changes which might help to improve the text and the discussion. In particular:

Abstract - 

lines 35-39 are a bit confusing: “Visualization by cryo-EM shows viral envelope disruption similar to what has been observed with AMP activity against other viruses. This suggests a common or biomimetic mechanism, possibly due to the differences between the phospholipid head group makeup of viral envelopes and host cell membranes. Furthermore, we observed no cytotoxicity against primary cultures of oral epithelial cells,.....

I believe that the authors want to say that the differences in the phospholipid head group makeup between the viral and host cells membranes might explain why these compounds destabilize the viral envelope while being not cytotoxic. In my opinion the second sentence should go after the third.

Line 107 

Figure 1B is cited but Figure 1A is never cited in the manuscript.

Figure 1B - 

The size of formulas is not homogeneous. Some chains look longer than others while they are of the same size. For example it seems that MXB-10 has a longer chain than MXB-5 but this is not the case. Please make all formulas of the same size for better comparison. 

Figure 2A - 

How do the authors explain that the presence of MXB-10 actually increases the viral DNA levels compared to control? Please discuss this point. 

Why is MXB-6 not shown in Figure 2? Please explain in the text or remove it from Figure 1B.

The activity essays in Figure 2 are reported in microgram per milliliter but the molecular formulas can have a quite different molecular weight (among them and in comparison to LL-37). In micromolar units MXB-1 could display almost double activity and LL-37 could be much more active than the peptoids. Please comment on that.

Figure 3B

For clarity, please specify somewhere in the figure that the two rows correspond to different images of the same experiment.

Line 231. I am not a native english speaker but shouldn’t “between” be replaced by “among”?

Lines 234-241. I do not see the evidence of the importance of halogenation. MXB-10 has two bromine atoms and it looks totally inactive. Out of the three chosen compounds (MXB-4,5 and 9), only 2 have halogens. More interesting is the comparison MXB9-MXB10. They only differ in the length of the long chain but one is very active and the other is inactive (in Figure 2A it actually seems to promote the infection). One could hypothesize that the length of the apolar chain does not adapt to the thickness of the bilayer. However the same long “inactivating” chain is present in the active compound 5. Please improve this part of the discussion.

In conclusion, this is a very interesting article clearly presenting the effect of peptoids on the envelope of HSV-1 and SARS-CoV-2. The subject is of high impact. In many ways I found it excellent and I think it should be published. However, it is important that the authors improve the discussion with a deeper insight in the part relating antiviral activity with molecular structure, to make it more convincing (or they repeat this experimental part to confirm the results).

Author Response

Abstract - 

lines 35-39 are a bit confusing: “Visualization by cryo-EM shows viral envelope disruption similar to what has been observed with AMP activity against other viruses. This suggests a common or biomimetic mechanism, possibly due to the differences between the phospholipid head group makeup of viral envelopes and host cell membranes. Furthermore, we observed no cytotoxicity against primary cultures of oral epithelial cells,.....

I believe that the authors want to say that the differences in the phospholipid head group makeup between the viral and host cells membranes might explain why these compounds destabilize the viral envelope while being not cytotoxic. In my opinion the second sentence should go after the third.

            Response: We have rearranged the sentences as suggested.

Line 107 

Figure 1B is cited but Figure 1A is never cited in the manuscript.

Response: This is now cited, after moving the rationale section to the introduction.

Figure 1B - 

The size of formulas is not homogeneous. Some chains look longer than others while they are of the same size. For example it seems that MXB-10 has a longer chain than MXB-5 but this is not the case. Please make all formulas of the same size for better comparison. 

            Response: The figure has been amended so that the chains are the same size.

Figure 2A - 

How do the authors explain that the presence of MXB-10 actually increases the viral DNA levels compared to control? Please discuss this point. 

            Response: We have addressed this in the results and discussion sections.

Why is MXB-6 not shown in Figure 2? Please explain in the text or remove it from Figure 1B.

            Response: We have now explained this in the text

The activity essays in Figure 2 are reported in microgram per milliliter but the molecular formulas can have a quite different molecular weight (among them and in comparison to LL-37). In micromolar units MXB-1 could display almost double activity and LL-37 could be much more active than the peptoids. Please comment on that.

            Response: We have now provided the concentration in micromolar units for the appropriate peptoids

Figure 3B

For clarity, please specify somewhere in the figure that the two rows correspond to different images of the same experiment.

            Response: We have now added text to explain this.

Line 231. I am not a native english speaker but shouldn’t “between” be replaced by “among”?

            Response: This has been amended to “among”.

Lines 234-241. I do not see the evidence of the importance of halogenation. MXB-10 has two bromine atoms and it looks totally inactive. Out of the three chosen compounds (MXB-4,5 and 9), only 2 have halogens. More interesting is the comparison MXB9-MXB10. They only differ in the length of the long chain but one is very active and the other is inactive (in Figure 2A it actually seems to promote the infection). One could hypothesize that the length of the apolar chain does not adapt to the thickness of the bilayer. However the same long “inactivating” chain is present in the active compound 5. Please improve this part of the discussion.

           Response:  This has now been discussed at length.